# Development of a decision support system for tsunami evacuation: application to the Jiyang District of Sanya City in China

Jingming Hou[1, 2, 3, 4], Ye Yuan[1, 2], Peitao Wang[1, 2], Zhiyuan Ren[1], Xiaojuan Li[3, 4]

[1] National Marine Environmental Forecasting Center, Beijing 100081, China
[2] Key Laboratory of Research on Marine Hazards Forecasting, State Oceanic Administration, Beijing 100081, China
[3] College of Resource Environment and Tourism, Capital Normal University, Beijing 100048, China
[4] Beijing Laboratory of Water Resource Security, Capital Normal University, Beijing 100048, China

Correspondence to: Jingming Hou (houjingming1982@126.com)

**Abstract:** Major tsunami disasters often cause great damage in the first few hours following an earthquake. The possible severity of such events requires preparations to prevent tsunami disasters or mitigate them. This paper is an attempt to develop a decision support system for rapid tsunami evacuation for local decision makers. Based on the numerical results of tsunami disasters, this system can quickly obtain the tsunami inundation and travel time from a numerical results database. Because numerical models are calculated in advance, this system can reduce decision-making time. Population distribution, as a vulnerability factor, was analyzed to identify areas of high risk for tsunami disasters. Combined with spatial data, this system can comprehensively analyze the dynamic and static evacuation process and identify problems that negatively impact evacuation, thus supporting the decision-making for tsunami evacuation in high-risk areas. When an earthquake and tsunami occur, this system can rapidly obtain the tsunami inundation and travel time and provide information to assist with tsunami evacuation operations.

**Keywords:** development, tsunami, evacuation, decision support system

## 1 Introduction

Tsunami can cause some of the worst marine disasters possible, and they affect many coastal countries around the world. Since the beginning of the 21st century, there have been a large number of global tsunami disasters, and over the last decade, two to three tsunami have occurred every year in the Pacific (IOC, 2013). Although tsunami are low-probability events, they are often accompanied by huge economic property loss and casualties (Papathoma et al., 2003). In addition, tsunami can cross oceans and influence areas far from their sources, resulting in large-scale disasters (Hébert et al., 2001). Following the 2004 Indian Ocean tsunami (Suppasri et al., 2011) and the 2011 Japanese tsunami (Wei et al., 2011), many governments and international organizations around the world have increased research into tsunami disaster prevention and mitigation. As an important part of tsunami mitigation, tsunami risk assessment is listed as a primary focus for tsunami disaster prevention and mitigation by the United Nations Intergovernmental Oceanographic Commission (IOC, 2015).

Tsunami risk assessments undertaken prior to the arrival of a tsunami are considered important and necessary (Sato et al., 2003; Strunz et al., 2011; Kurowski et al., 2011). According to natural

disaster risk assessment theory, risk assessment provides a means to quantify risk by analyzing
potential hazards and evaluating vulnerability conditions. Tsunami evacuation research has been
conducted in high-risk areas around the world, based on evaluations of tsunami hazard, vulnerability
and risk assessments. When a tsunami disaster happens, the first task for emergency rescue personnel is
to evacuate people to safe areas (Mück, 2008). Tsunami evacuation research needs related information,
such as the degree of hazard, and the distribution of people, roads, and shelters (Scheer et al., 2011).
Currently, many governments (e.g., Japan, the United States, and Thailand) issue tsunami evacuation
plans (Scheer et al., 2011) to help citizens plan for tsunami disasters.
This paper develops a decision support system for local decision makers to facilitate the planning
of tsunami evacuations and evacuation practices. Before the tsunami, the support system can analyze
the tsunami evacuation by retrieving the tsunami hazard and simulating the evacuation to identify
possible problems in the evacuation process. According to the analysis results, local decision makers
can take measures such as traffic control and widened lanes to improve the evacuation operations.
When an earthquake and tsunami occur, the support system can quickly provide the required
information for appropriate recommendations and decision-making to aid evacuation.
**2 Methods**
In the last 20 years, static and dynamic approaches have been used in tsunami evacuation research.
The static approaches have included the least-cost-distance model (Wood and Schmidtlein, 2013),
genetic algorithms (Park et al. 2012), and discrete element methods (Abustan et al. 2012). Several
dynamic approaches can also be found in the literature, e.g., traffic simulation models (Naghawi and
Wolshon, 2010) and agent-based models (Mas et al. 2015).
The static least-cost-distance model is suitable for tsunami evacuation planning over relatively
large areas focusing on finding the shortest path from the hazard zone to a safe location, whereas the
dynamic agent-based model was developed for a localized area focusing on the evacuees' behavior and
dynamic travel costs. We adopt a combination of least-cost-distance and agent-based models in our
support system.

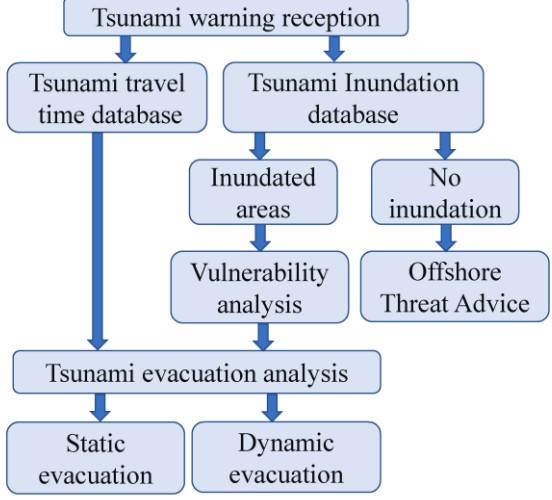


**Figure 1. Framework of the decision support system**

A framework for this system is shown in Fig. 1. The system aims to gather all the information for tsunami hazards and vulnerability, and analyze the tsunami evacuation for a comprehensive evacuation plan and evacuation practices. This system development process has three stages: (1) hazard analysis, (2) vulnerability analysis, and (3) evacuation analysis. All potential tsunami are simulated by numerical models using a range of possible magnitudes: 7.0, 7.5, 8.0, 8.5, and 9.0. The calculation results, including tsunami travel time and tsunami inundation, are then imported into a database. If there is no tsunami inundation, offshore threat advice is given; if there is a tsunami inundation, the tsunami evacuation is analyzed with the tsunami travel time in the database and vulnerability analysis. The vulnerability of the inundated region is investigated and high-risk areas are identified. Vulnerability mainly analyzes the population distribution to ensure the proper evacuation of the entire population. Then, both the static evacuation analysis and dynamic evacuation analysis are conducted for future policy-making and evacuation practices.

It may take 2 minutes to get tsunami hazard information using the decision support system. The vulnerability analysis requires 2 to 4 minutes. It takes 5 minutes for the system to provide tsunami static analysis results. Tsunami dynamic evacuation analysis may take several hours, but this analysis can be used to study local tsunami evacuation problems before a tsunami occurs.

Suitable data are the basis of tsunami evacuation analysis. To build this evacuation system, data as shown in Table 1 are used.

**Table 1. Data required for the decision support system**

| No. | Data types | Purposes |
|---|---|---|
| 1 | Tsunami travel time | Hazard analysis |
| 2 | Tsunami inundation | Hazard analysis |
| 3 | Population distribution | Vulnerability analysis |
| 4 | Population characteristics | Evacuation analysis |
| 5 | Elevation data | Evacuation analysis |
| 6 | Land use data | Evacuation analysis |
| 7 | Evacuation shelters | Evacuation analysis |
| 8 | Street network | Evacuation analysis |

**3 Overview of the study area**

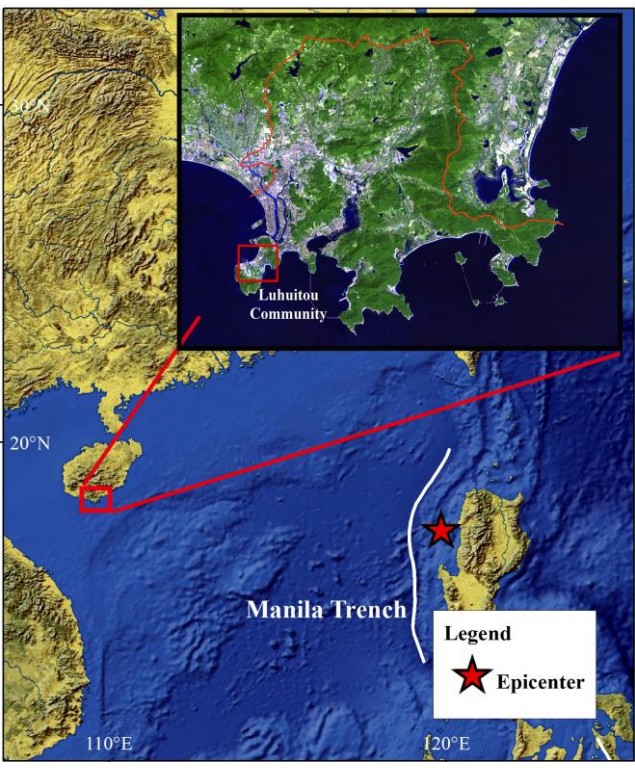


**Figure 2. Location map of the study area**

This system is applied to the Jiyang District of Sanya City in China to show the development of
the system for tsunami evacuation. Sanya, located at the southern tip of Hainan Island, is a
transportation and communication hub in southern Hainan Province and an important port on the
southeast coast of China. Sanya has a unique geographical advantage in international economic
relations as it lies close to many ASEAN countries. The Jiyang District (Fig. 2) lies in the heart of
Sanya City. The topography is high in the north and low in the south. Most of this area's elevation lies
between 10 and 300 m, while the maximum elevation is 604 m. Plains dominate the terrain in this
coastal area. An 8.5 earthquake in the Manila Trench is assumed to be the tsunami source (Fig. 2).

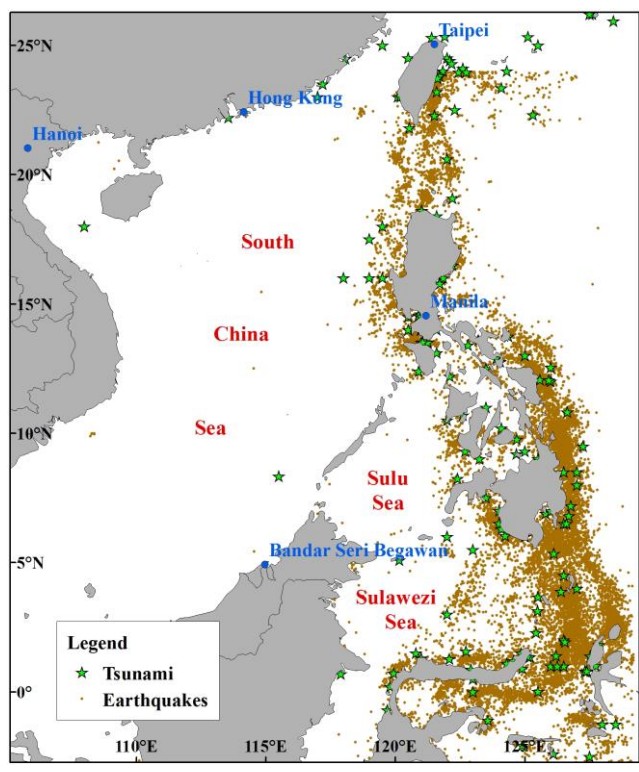


**Figure 3. Location of the South China Sea**

As shown in Fig. 3, the potentially most dangerous tsunami for the Jiyang District originate from the South China Sea region. The South China Sea, located in the western Pacific Ocean, and covering an area of about $3.5 \times 10^6$ km$^2$, is one of the largest marginal seas in East Asia (Liu et al., 2007).

The Manila Trench, an active subduction system where the Eurasian Plate subducts beneath the Philippine Plate, lies in this region. The Manila and Sulawesi subduction zones have been identified as potential tsunami sources by the United States Geological Survey (USGS; Kirby, 2006). Written records can be found of historical tsunami in the northeastern South China Sea (Lau et al., 2010; Megawati et al., 2009). For example, Sun et al. (2013) reported preliminary evidence from the Xisha Islands in the South China Sea for a large tsunami around AD 1024.

Earthquake records may underestimate the tsunami potential of the region (Okal et al., 2011). Although the probability of a major earthquake in the Manila Trench is not high, this does not mean that a major earthquake will not occur in the future (Megawati et al., 2009). Dao et al. (2009) simulated a worst-case tsunami scenario for the Manila Trench using a numerical model, and estimated a tsunami height of 14 m in the vicinity of the Philippines and southwest of Taiwan. A tsunami triggered by a giant earthquake from the Manila Trench could cause devastating damage to the Philippines, southern China, and Vietnam (Megawati et al., 2009; Ren et al., 2015).

In this paper, we consider a number of potential tsunami scenarios in the South China Sea region. The historical earthquakes and tsunami from the region are shown in Fig. 3. Earthquake data (1976–2016) are from the USGS, whereas the tsunami data (2000–2016) are from the US National Geophysical Data Center (World Data System, 2016). These data include event times, locations, and types.

## 4 Tsunami hazard analysis

A database was used in this decision support system to quickly determine the tsunami hazard. All the historical tsunami sources, covering a wide range of magnitudes, were simulated by numerical modeling in advance, including the tsunami inundation and tsunami travel time, which were then stored in the database. Once a tsunami is triggered, the system can retrieve the tsunami travel time and inundation areas from the database. The seismic source parameters for hypothetical earthquakes with different magnitudes are shown in Table 2 (Igarashi, 2013). The nodal plane parameters for the moment tensor are based on Slab 1.0 (Hayes et al., 2012). An 8.5 magnitude earthquake was used as a case study to demonstrate the system.

**Table 2. Fault parameters for a number of hypothetical earthquakes**

| Magnitude | Rupture length (km) | Rupture width (km) | Strike (°) | Dip (°) | Slip (°) | Focal depth (km) |
|-----------|--------------------|--------------------|-----------|---------|----------|------------------|
| 9.0 | 398.1 | 199.1 | 1 | 41 | 70 | 24.6 |
| 8.5 | 223.9 | 111.9 | 1 | 41 | 70 | 24.6 |
| 8.0 | 125.9 | 62.9 | 1 | 41 | 70 | 24.6 |
| 7.5 | 70.8 | 35.4 | 1 | 41 | 70 | 24.6 |
| 7.0 | 39.8 | 19.9 | 1 | 41 | 70 | 24.6 |

### 4.1 Tsunami inundation

A multi-grid coupled tsunami model (COMCOT) was used to calculate the tsunami wave amplitude. This model was developed at Cornell University (Liu et al. 1998), and has been used to successfully simulate several historical tsunami events (Wang and Liu 2006). It can study the entire life-span of a tsunami, including the inundation.

When the tsunami propagates over the continental shelf, linear shallow-water equations are no longer suitable. The tsunami wavelength becomes shorter and the amplitude increases. The significance of the Coriolis force and the frequency dispersion decreases. The COMCOT model uses nonlinear shallow-water equations, including the bottom friction terms, to simulate the tsunami in the coastal zone.

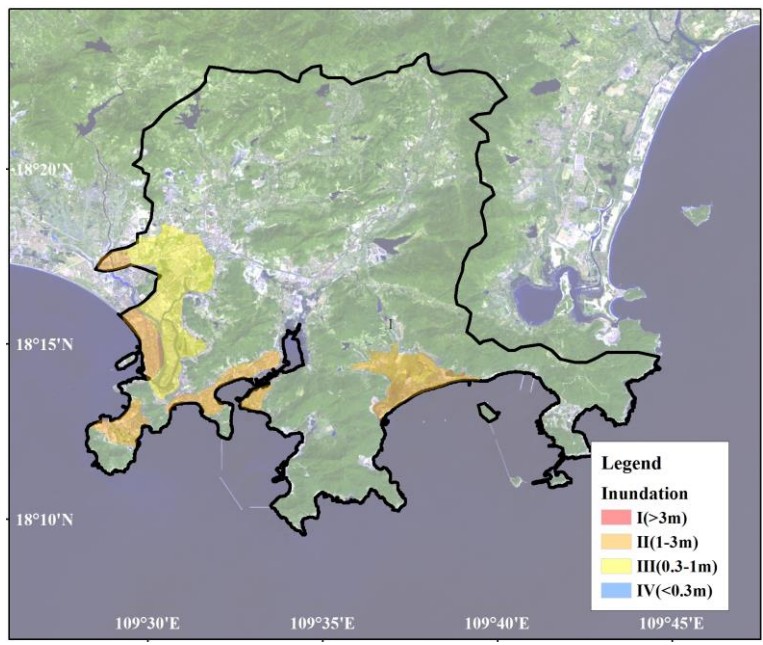


**Figure 4. Inundation in the Jiyang District**

The inundation caused by the 8.5 magnitude earthquake tsunami covered an area of 29 km$^2$ (Fig.
4). The inundation areas were classified into level 2 and level 3; the maximum amplitude is 1.8 meters.
There are hospitals, schools, shopping malls, and hotels in the inundation areas. Several communities
such as Luhuitou, Yalongwan, Hexi, Yulin, and Anyou were inundated.
**4.2 Tsunami travel time**
Tsunami travel time is a very important factor for tsunami evacuation systems. Evacuation time is
the available response time for evacuation (Post et al., 2009), less than the tsunami travel time. The
type of evacuation method that should be adopted (horizontal or vertical evacuation) mainly depends
on the amount of evacuation time available.
The evacuation time consists of four components (Post et. al., 2009): Estimated Tsunami Arrival
time (ETA), Institutional Decision Time (IDT), Institutional Notification Time (INT), and Reaction
Time (RT) of the population. The available response time for evacuation (RsT) can be obtained from:

RsT = ETA – IDT - INT - RT          (1)

In this system, the tsunami travel time was calculated by the tsunami travel time model (TTT),
developed by Paul Wessel for Geoware. This model is an application of Huygen's principle with water
depth as the only variable (Murty, 1977) and is based on Eq. (2). In Eq. (2), $h$ denotes water depth, and
$C$ represents tsunami velocity.
$$C = \sqrt{gh}. \qquad\qquad (2)$$




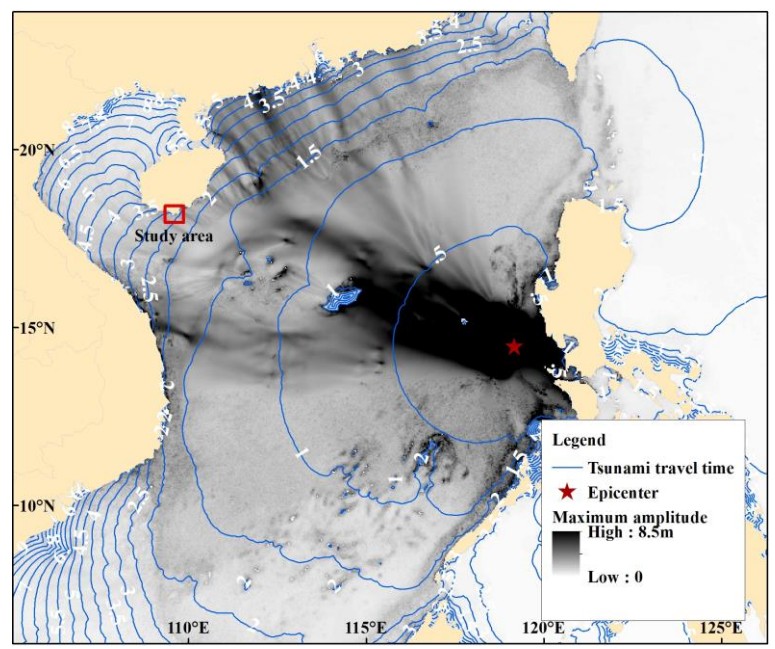

**Figure 5. Tsunami travel time and maximum amplitude**

The tsunami travel time and maximum amplitude of an 8.5 earthquake tsunami in the Manila Trench, obtained from the hazard database, are shown in Fig. 5. According to this figure, the tsunami arrival time for the study area is approximately 2.9 h.

By considering the emergency response capability of local authorities, it was found that the total time for tsunami warning notification, reception, and public response would be 0.8 h. It was calculated that the exposed population would evacuate in 2.1 h.

**5 Vulnerability analysis**

The second stage component in the system is vulnerability analysis. The vulnerability analysis follows the hazard analysis to specify the number of people at risk. The population factor is used to measure the vulnerability of the inundation areas and identify areas of high risk. Vulnerability analysis results also can help government to enhance vulnerability management and reduce the level of vulnerability.

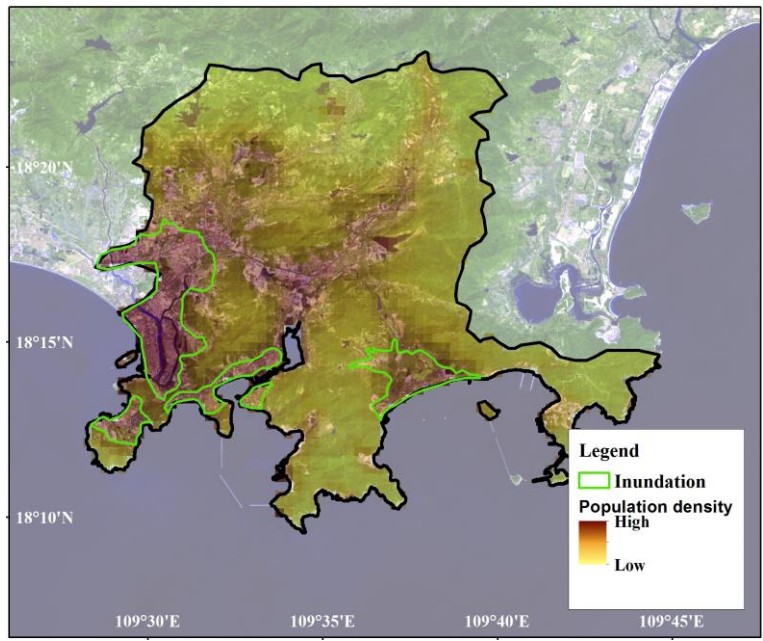

**Figure 6. Vulnerability map based on population density**
Population data can be used to identify where evacuations should take place and what kind of
evacuation measures should be adopted. Such information will help decision makers develop measures
to mitigate tsunami disasters (National Research Council 2007). Although there is agreement that there
are no specific variables that can describe the social vulnerability of people, there are some indicators
that are often used to represent vulnerability, such as age, race, gender, nighttime and daytime
population density, income, and special needs of the population (Alabdouli 2015). The locations and
numbers of people in inundation zones can be identified by superimposing hazard and demographic
data (Wood and Schmidtlein 2013). Our system analyzes population distributions from recent
population census data. Jiyang District has a population of about 240,000, and covers an area of 372
km$^2$. The population distribution of Jiyang District and inundation areas are shown in Fig. 6, and it can
be seen that some densely populated areas of Jiyang District are located in inundation areas. There are
114,086 people in the inundation areas, which cover an area of 29 km2. 52% of the population is male
and 48% is female. The age distribution of the people age is 15% elderly (65 years old and over), 65%
middle-aged (18–65 years old), and 20% young (less than 18 years old).
Using vulnerability analysis, several high-risk areas were identified. The evacuation analysis was
conducted mainly in the high-risk areas.
**6 Tsunami evacuation analysis**
Evacuation, with the purpose of saving lives, plays a crucial role in tsunami disaster mitigation
plans. NTHMP (2001) indicated that the primary strategy of tsunami disaster mitigation is to evacuate
people from the hazard zone as quickly as possible – and before the tsunami arrives. There are two
main evacuation methods: horizontal evacuations and vertical evacuations. The former method
evacuates people away from the coast to safe zones (that may have a higher elevation, such as a hill);
the latter approach evacuates people to nearby tsunami-resistant buildings.
When managers make a decision about whether to evacuate and how to evacuate, they need

current information on elevation, roads, and shelters. In this study, the evacuation analysis was conducted in the Jiyang District and for the Luhuitou Community, one of the high-risk areas. Evacuation analysis included static and dynamic analysis. The static evacuation analysis mainly investigated evacuation cost, road congestion, and areas serviced by shelters, while the dynamic evacuation was analyzed using the agent-based model.

**6.1 Static evacuation analysis**

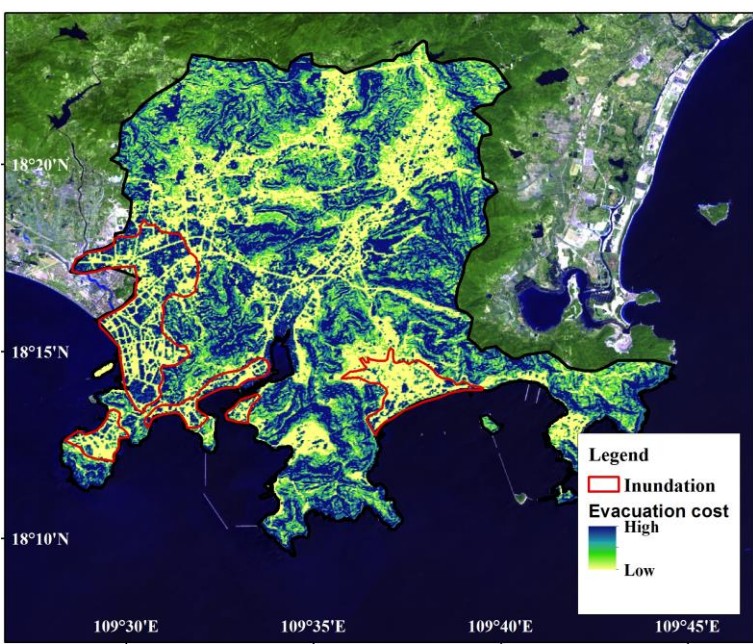

**Figure 7. Evacuation cost map based on land use and slope**

The environment that people evacuate from can influence the efficiency of evacuations. The evacuation cost should therefore be analyzed to provide information and suggestions to help managers make appropriate decisions (Sugimoto et al., 2003).

The best evacuation routes are not always short straight lines. Different types of land use can impact evacuations. For instance, evacuation by road is significantly faster than across agricultural land. An evacuation cost raster, which quantifies the degree of difficulty for each cell in an evacuation area, can be used to compute the influence of different environments on evacuation routes (ADPC, 2007). Evacuation cost is a combination of land use and slope. The best evacuation route should correspond to the lowest evacuation cost area.

Evacuation costs were analyzed in this system according to the approach of Wood and Schmidtlein (2013). The spatial analysis of ArcGIS was used to create a cost surface raster that considers the difficulty of evacuation through each cell. Each cell of this raster represents an inverse speed required for evacuation. Fig. 7 shows the evacuation cost as applied to the Jiyang District. The evacuation cost of the yellow area is relatively low. The evacuation routes should be designed in the low-cost area.

If there is enough evacuation time, people should be able to evacuate horizontally to a safe area outside the tsunami inundation zone after receiving the tsunami warning. If there is not enough time to evacuate, people can evacuate vertically to higher terrain or structures near the coast (Heintz and Mahoney, 2012). When a tsunami occurs, decision makers need to know whether or not there is road

congestion and where to evacuate to before inundation starts. Our decision support system provides congestion-prone road analyses to support evacuation actions.

Road condition is a very important factor in both horizontal and vertical evacuations, as the evacuation of a large number of people in a short time may lead to road congestion, especially in a populous coastal city. For example, on April 11, 2012, thousands of people were stuck in traffic congestion after a tsunami warning was issued in Indonesia (Wu et al. 2015). Evacuation managers need to know which roads are prone to congestion. In our decision support system, congestion-prone roads (Fig. 8) were analyzed by overlying population census data and road classification data. Roads are easily congested in places where there is a larger populace. There are 240,000 people in the Jiyang District, and the built-up area covers 61 km$^2$, which equates to 3934 people per km$^2$. It is easy to cause congestion when people receive a warning and begin to evacuate. From the results of the congestion-prone roads we can see that some of the roads in the inundated areas are congestion-prone roads in high level, which need more attention.

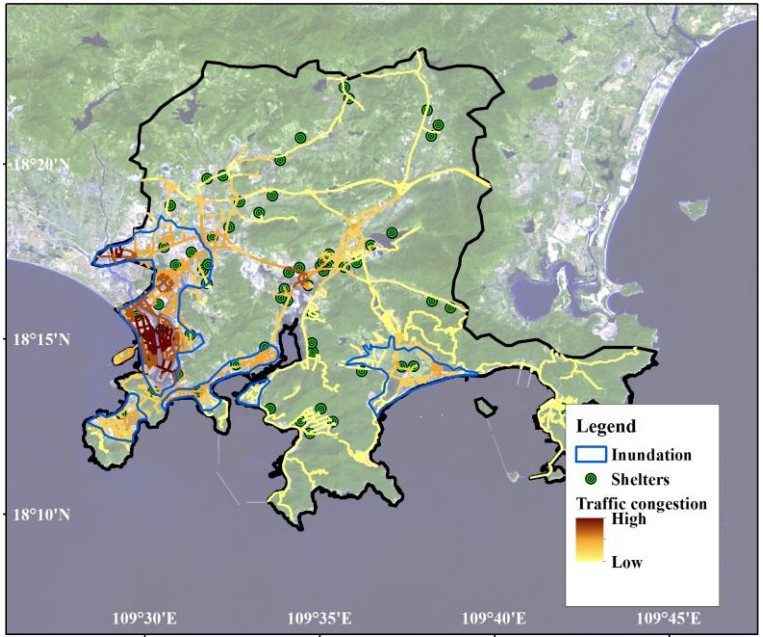

**Figure 8. Congestion-prone roads based on population census data**

Evacuation shelter buildings provide a destination for an evacuation. At the beginning of the evacuation analysis, some shelters (Fig. 8) are provided by the local managers. New shelters can be added according to the need in the system. Evacuation shelters are usually selected in lower cost and higher-lying areas, preferably with a road connection. In addition, the structure, function, and security of candidate shelters should also be investigated (Budiarjo, 2006). Generally, places with social functions are often used as shelters, e.g., schools, hospitals, shopping malls, convention centers, stadiums, hotels, and parks.

Each shelter responds to the evacuation of several communities, and the service area of each shelter was defined in our system. The cost allocation method in GIS was used to analyze the service area. This method uses the least accumulative cost to calculate the nearest source in the evacuation cost grid. The service areas of vertical shelters of the Luhuitou Community, one of the high-risk areas, are shown in Fig. 9.

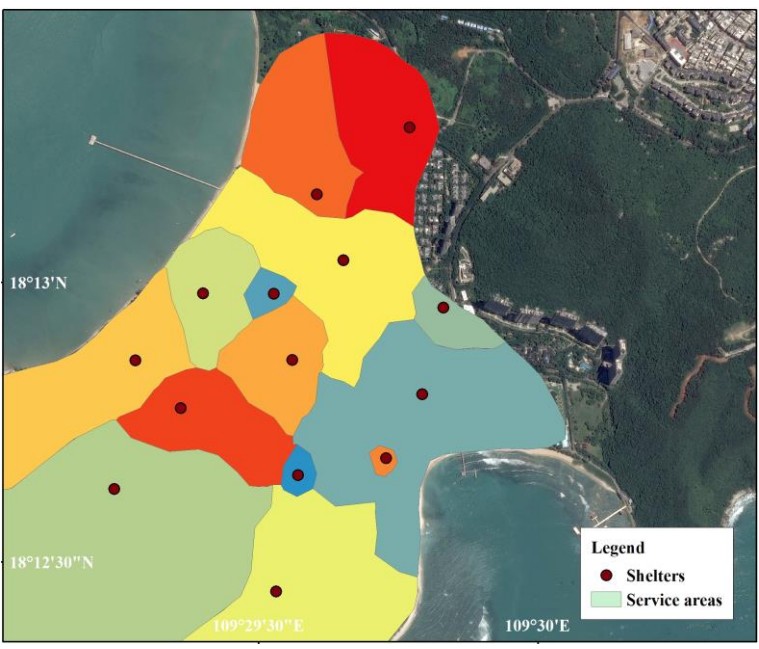

**Figure 9. Service areas of vertical shelters for the Luhuitou Community**

To reach the shelters, residents would need the main horizontal evacuation routes and bus pickup locations. As shown in Fig. 10, our decision support system can show the main evacuation routes, shelters, and bus pickup stations for the Luhuitou Community. The locations of shelters and bus pickups can be adjusted according to the different inundation areas.

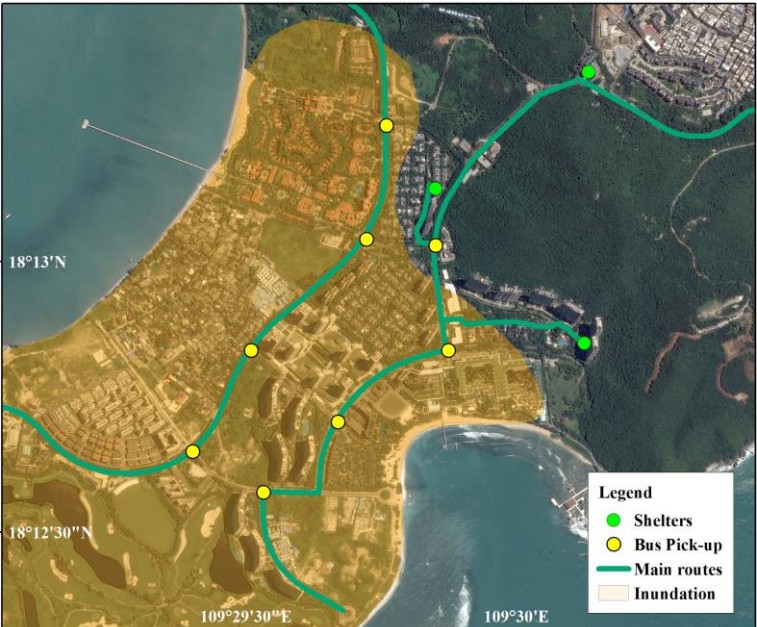


**Figure 10. Pickup locations and main horizontal evacuation routes for the Luhuitou Community**


**6.2 Dynamic evacuation analysis**

The static evacuation analysis attempts to find the shortest path for horizontal and vertical
evacuation, without considering the dynamic changes in the process of evacuation. However, the
agent-based model can simulate the dynamic interactions and actions of the autonomous agents, and
focuses on the evacuees' behavior.
This study used the NetLogo environment (Wilensky, 1999) to conduct the dynamic evacuation
analysis. NetLogo uses an agent-based modeling approach to simulate the interactions and actions of
all autonomous agents. Each agent assesses their situation and makes evacuation decisions based on
specified rules. Evacuation modeling results can show (1) the mortality rate, (2) the provision of
vertical evacuation shelters, (3) the congestion and bottlenecks of road networks, and (4) the choice of
vertical and horizontal evacuation. The percentage of evacuation by car in this case is from 0 to 100%.
Walking speed is assigned from 1 m/s to 3 m/s depending on different ages (Wang et al., 2015).

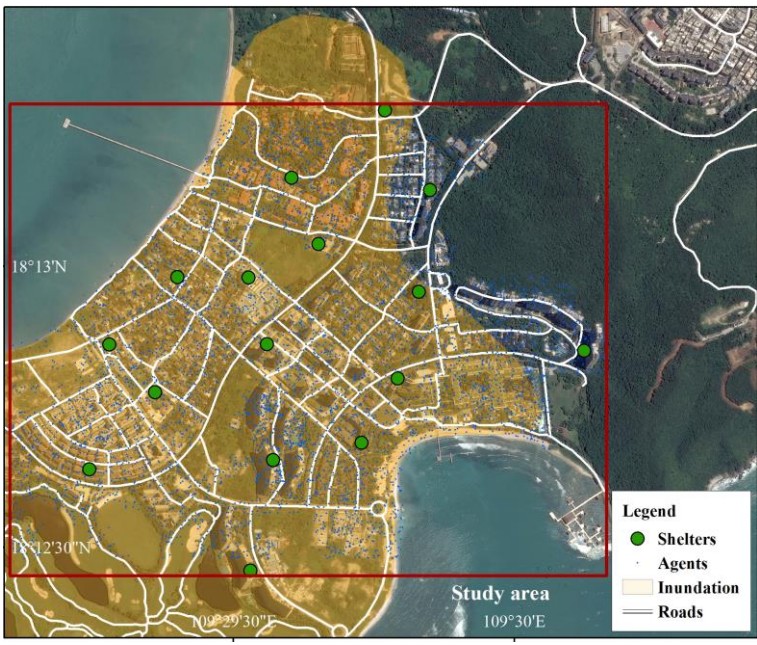

**Figure 11. Study area for dynamic evacuation analysis**

Fig. 11 shows the study area for dynamic evacuation analysis of the Luhuitou Community. The
dynamic evacuation analysis results revealed useful information regarding evacuation. Several
bottlenecks (Fig. 12) were identified as a result of the use of vehicles in this area. With the increase in
car use, the mortality rate ascends because of traffic congestion. The number of vertical evacuation
shelters is very effective in reducing mortality. Based on these results, emergency managers can take
appropriate measures to improve evacuation operations.

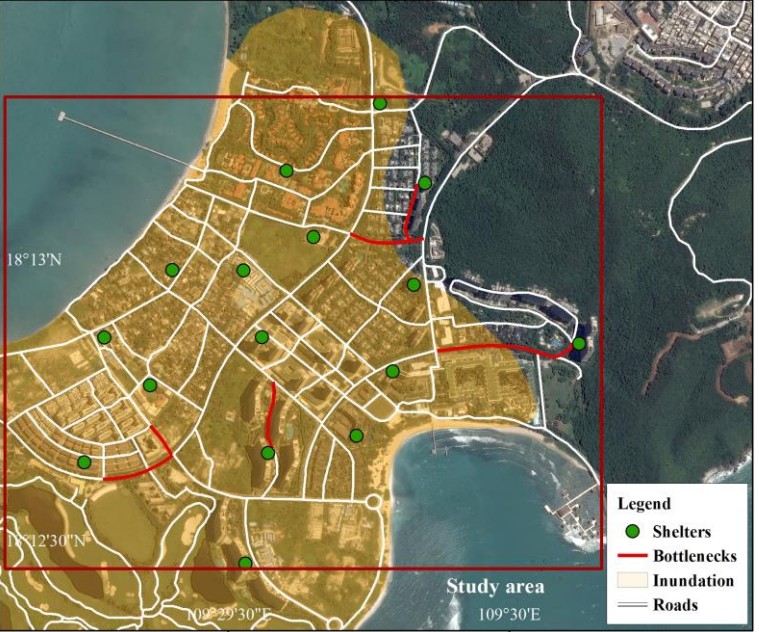


**Figure 12. Bottlenecks identified from dynamic evacuation analysis**
**7 Conclusion**
Tsunami evacuation research involves not only tsunami hazard factors, such as tsunami travel
time and inundations, but also vulnerability factors like population distributions. The selection of an
appropriate evacuation method depends on the range of possible geographical environments and
evacuation times. Evacuation decision makers need a variety of information to direct evacuation
actions appropriately.
Based on the understanding of a future tsunami hazard, this paper presented a decision support
system for tsunami evacuation of tsunami-prone areas. The presented system considers tsunami hazard,
vulnerability, and evacuation analysis, with the purpose of helping prepare for disaster risk
management and evacuation planning in the tsunami-prone areas.
The development of this decision support system requires a variety of geographic data, including
catalogs of historic earthquakes and tsunami, water depth, digital elevation models, satellite images,
evacuation shelters, and roads. Note that the tsunami risk of a certain region should be assessed roughly
before the development of this system. The system is best developed in an area that is likely to suffer a
future tsunami disaster.
When there is no tsunami, this system can be used to simulate the evacuation and identify
potential problems influencing the evacuation. Based on the simulation results, local managers can take
measures to mitigate the adverse factors and make evacuation plans. Once a tsunami occurs, this
system can be used for interactive and adaptive evacuation management.
The decision support system has been applied to the Jiyang District of Sanya City, China. A total
area of 29 km$^2$ areas was inundated by an 8.5 earthquake tsunami that had its origins near the Manila
Trench. Some bottlenecks were found that would affect the Luhuitou Community in a tsunami
evacuation. The analysis results can help local managers better understand the tsunami hazard and take
measures to improve the evacuation status. For example, vulnerable buildings should not be built in
inundation areas and road networks could be improved to mitigate congestion. However, the decision
support system needs to be further modified and improved. Future research will focus on
comprehensive evacuation simulation by incorporating a vulnerability analysis of major facilities, and
other advanced evacuation models.
**Acknowledgements**
This work has been funded by The National Key Research and Development Program of China
(Grant Nos. 2016YFC1402000 and 2016YFC1401500) and The Chinese Public Science and
Technology Research Funds Ocean Projects (Grant No. 201405026).

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
