# Peer review of "Development of a decision support system for tsunami evacuation: application to the Jiyang District of Sanya City in China"

_Natural Hazards and Earth System Sciences, 2016_

## Referee Comment (RC1) · Anonymous Referee #1 · 8 Dec 2016

Reviewer Comments for Manuscript Number nhess-2016-319

Title: Development of a decision support system for tsunami evacuation in the South China Sea region

Authors: Jingming Hou, Ye Yuan, Peitao Wang, Zhiyuan Ren, and Xiaojuan Li

Manuscript Type: Research article

Dear Editor and Authors,

General comments:

Firstly, thank you for submitting this paper and giving me the opportunity to review it.

[Figure]

Please, consider all comments and suggestions simply as the opinion of this reviewer.

This document addresses the development of a decision support system for tsunami evacuation and its application to the Jiyang District of Sanaya City, located at the southern tip of Hainan Island (South China region).

From the reviewer's point of view, both the tsunami evacuation itself and the development of an evacuation-related decision support system are topics of great interest since this is a relevant key issue related to tsunami risk for communities affected or potentially affected by this threat.

However, the general comment for the whole paper is that the reviewer has not been able to find any significant point regarding the principal criteria of the reviewing process (scientific significance, scientific quality and presentation quality).

There is no clarity in all the process initially described, no description of how each one of the analyses have been performed (there is no possibility for replicability) there is no description of results either and almost no conclusions.

I think that there is no connection between the different parts of the process. Apparently, one feed each other but this is not shown throughout the document.

Considering the above mentioned and after reflection, the final consideration for the review is: major revisions.

Below, there are also some specific comments intended to contribute to the improvement of the article, but the general recommendation is to "rethink" the document.

Specific comments:

Lines 61-62 It is the reviewer opinion that agent based model is not developed for tsunami drills. There may be relation between AB model and tsunami drills but it is not the "objective" of the modelling.

Lines 62-66:

[Figure]

It seems that this a justification for choosing LCD model but I would recommend to consider the revision of these sentences. The idea is not clear.

Figure 1: The population distribution is not included in the framework.

Line 73: I think that "effected" should be replaced by "affected"

Lines 73-74: The same idea is repeated in two consecutive sentences.

Lines 89-90: It is difficult to see the coherence between the idea of modern earthquake records overlooking the tsunami potential and that "Modern seismic analysis suggests that the 1918 Morro Bay and 1934 Luzon earthquakes were larger than their officially reported magnitudes".

Lines 97-101: Is not clear what else is done after considering a "number of potential tsunami scenarios..." Are they simulated? Are they collected?

Figure 3: This seems to be the location map of the study area. However the study area is not clear (scale is very small) and there is that green line and the epicenter. I would recommend to include a figure with much more detail of the study area and the other one (together with figure 4, for example) related to the travel time calculations.

Lines 117-119: This is not a formal definition of evacuation time, but the interpretation of the authors. I miss either saying that the definition provided is how it is understood in this paper or the reference that has been taken.

Lines 117-130: The calculation performed to establish the tsunami travel times is not clearly described. I believe that the method should be replicable but any other person reading the document: for instance, How is the average water depth determined? epicenter coordinates? Results (2 hours) may be good as a rough approximation but I do not believe they are comparable (at least in general terms) with numerical tsunami travel time models. The approximation will be acceptable depending also on the purpose followed. Maybe for evacuation purposes in a decision support system in a local area this is not the best. On the other hand, it could be debatable if average velocity is

appropriate to estimate the time. . .

Lines 147-195: From this reviewer's point of view, vulnerability is not really being considered and is not properly defined. Maybe it could reshaped on some way explaining that some aspects that could influence in the vulnerability of the system are taken into account. . .

There is confusion in the statement of lines 151-152

Paragraphs following the line 156 should follow the order detailed in the sentence, for an easier reading.

Line 160 and Line 168: I really believe that elevation is not a preferred consideration for vulnerability analysis. It is a crucial parameter to establish the affected area but that is hazard and exposure.

Lines 186 to 189. Is the "8 km distance from shore" calculated based on the previous equation? This is not clearly stated.

Lines 192 to 195: It is true but has it been done in the frame of this work? If yes, please show results. If not, clarify it. Is really the vulnerability being calculated or estimated in some way? Is there any result achieve?

Line 218: This is not very relevant when describing land use to estimate the impedance related to evacuation analysis

Line 224: There is no explanation of the figure.

Lines 236-237: Congestion-prone roads are calculated based on population census and road classification data. Which is the approach followed? There is no explanation. It seems obvious than "worst roads" and most densely populated areas will have more congestion-prone roads, but it is no explained. Results are not explained either.

Lines 241-246: It is no clear if vertical shelters are added to the system based on calculations that are not the objective of this paper or if this have been addressed in this

work. It is stated, as the only apparent reason, that evacuation shelters are usually se-
lected "following the principles of good accessibility and large capacity, among others".
This is a quite poor reasoning. As figure 10 has not been explained, the reader is not
able to understand how those green points are calculated. On the other hand, if there
has not been inundation modelling, which is the procedure followed to locate them?

Conclusions: I think they should be remade after a careful revision of the article.

---

## Referee Comment (RC2) · Anonymous Referee #2 · 10 Dec 2016

Title of this manuscript sounds interesting but the detail seems to be lower than what have stated in the title. The manuscript needs major improvement in their method, study area, explanation and English writing so that it can be reached to international standard. Please find my comments as shown below.

Major comments - Only one study area in China cannot represent the evacuation for whole SCS region. If the authors would like to do such quick and simple method, at least they should have one or two more target areas to show performance of the system in both near-field and far-field tsunami. Verification of the model is needed to convince readers that even such simple method is applicable. - How fast is the processing time required for this decision support system? I suggest to add information of time in Fig.

1. I believe that each country in SCS region has different timing of tsunami warning preparation. Please add information about this and tell readers how this system can help at each timing for SCS countries. - The use of English should be very much improved by a native speaker.

Specific comments - Title: The title should be more specific by mentioning the study area. - L36-37: This reference Benard (2005) is too old to support the statement "Currently, many governments.." - L53: Objective and purpose of this paper should be written at the end of section 1. - L55-67: More literature review is needed. There are more types of evacuation model than what have mentioned in this part. I don't think that the agent-based model is only for evacuation drill. It can be for disaster planning as well. What do you mean by "the decision makes do not know well"? - L70-78: Should other items in Fig. 1 be explained? How much detail of fault mechanism considered in the database? Do they also have events outside of subduction zone in the database? Explain briefly about evacuation cost here. - L117-129: Evacuation time can also be a time from a natural warning (i.e. ground shaking) in case of no warning system. The authors should scientifically state why equation (1) is proper to be applied to this region. They should also more clear by saying that the tsunami travel time is estimated by dividing the distance from epicenter to target area by tsunami celerity, C (not to be confused with flow velocity). What do you mean by "numerical tsunami travel time model"? Fig. 4 should be improved by adding the location names (i.e. China, the Philippines, Hinan, Luzon, Manila Trench) and the estimated tsunami celerity during the deeper (4,000 m) and shallower (2,000 m) section in the figure - L133-142: "Influence area", what is the meaning of "influence"? Is it from tsunami amplitude, arrival time or any other parameters? The simple radius based on earthquake magnitude shown in Fig. 5 is too simple. As the authors had mentioned in section 4.2 that the evacuation time is calculated by the sea depth, the sea depth, coastal topography of each communities for both sides of the Philippines are quite different and cannot be just simply represent by the radius. - L161: How do you relate "magnitude (do you mean earthquake magnitude?)" to "physical nature of tsunami"? - L182-191: I am not

sure if the 8.3 km was really happened. Where was it and what kind of topography? Please scientifically explain why equation (2) is suitable to apply to your study area or other areas in SCS region. Only tsunami height is a parameter for equation (2). Such large overestimation can be occurred in mountain areas where tsunami is limited by the topography. The authors should also explain why they used the distance of 8 km as shown in Fig. 8. What is their expected Y0 to get 8 km distance? - L221-237: There are many parameters related to the traffic during evacuation such as road width, traffic regulation during tsunami warning, distribution of evacuation shelters, ratio of evacuation using car, number of people in one car, day or nighttime that the authors did not mention. Explanations of the evacuation cost analysis is too simple and not enough to understand. - L247-262: The conclusion is rather simple. They should write, for example, major finding in their study or benefit (from their discussed results) to agent-based model or suggest how to develop your proposed system to other countries or regions.

---

## Author Comment (AC1) · 9 Feb 2017

Authors' response to Anonymous Referee #1

First of all, we greatly appreciate Referee #1 for the valuable and constructive comments. We completely agree with the Referee. According to the comments made by the Referee, we rethink this paper and carry out an extensive revision of this paper. Our responses to the comments are shown below. The revised paper is in the supplement.

Referee #1

General comments:

This document addresses the development of a decision support system for tsunami evacuation and its application to the Jiyang District of Sanaya City, located at the southern tip of Hainan Island (South China region). From the reviewer's point of view, both the tsunami evacuation itself and the development of an evacuation-related decision support system are topics of great interest since this is a relevant key issue related to tsunami risk for communities affected or potentially affected by this threat. However, the general comment for the whole paper is that the reviewer has not been able to find any significant point regarding the principal criteria of the reviewing process (scientific significance, scientific quality and presentation quality). There is no clarity in all the process initially described, no description of how each one of the analyses have been performed (there is no possibility for replicability) there is no description of results either and almost no conclusions. I think that there is no connection between the different parts of the process. Apparently, one feed each other but this is not shown throughout the document. Considering the above mentioned and after reflection, the final consideration for the review is: major revisions. Below, there are also some specific comments intended to contribute to the improvement of the article, but the general recommendation is to "rethink" the document.

General Response:

We would like to thank this Referee for accepting to review this paper and for the valuable and constructive comments. We rethought the decision support system and modified the study framework. Please see our responses to the specific comments below. This paper develops a decision support system for local decision makers to facilitate the planning of tsunami evacuations and evacuation practices. Before the tsunami, the support system can analyze the tsunami evacuation by retrieving the tsunami hazard and simulating the evacuation to identify possible problems in the evacuation process. According to the analysis results, local decision makers can take measures such as traffic control and widened lanes to improve the evacuation operations. When an earthquake and tsunami occur, the support system can also quickly provide the required

information for appropriate recommendations and decision-making to aid evacuation.

Specific comments:

Comment (a):

Lines 61-62 It is the reviewer opinion that agent based model is not developed for tsunami drills. There may be relation between AB model and tsunami drills but it is not the "objective" of the modelling.

Response (a):

This has been modified in Lines 51-53 of the revised paper.

Comment (b):

Lines 62-66: It seems that this a justification for choosing LCD model but I would recommend to consider the revision of these sentences. The idea is not clear.

Response (b):

This has been clarified in Lines 51-54 of the revised paper. We adopt a combination of least-cost-distance and agent-based models in the revised paper.

Comment (c):

Figure 1: The population distribution is not included in the framework.

Response (c):

According to the comments of the Referees, we rethink this system. The new framework is shown in Line 56. The population factor is used to measure the vulnerability of the inundation areas and identify areas of high risk.

Comment (d):

Line 73: I think that "effected" should be replaced by "affected"

Response (d):

This sentence has been deleted because of the changed framework.

Comment (e):

Lines 73-74: The same idea is repeated in two consecutive sentences.

Response (e):

This has been modified in Line 63 of the revised paper.

Comment (f):

Lines 89-90: It is difficult to see the coherence between the idea of modern earthquake records overlooking the tsunami potential and that "Modern seismic analysis suggests that the 1918 Morro Bay and 1934 Luzon earthquakes were larger than their officially reported magnitudes".

Response (f):

This has been modified in Line 101 of the revised paper.

Comment (g):

Lines 97-101: Is not clear what else is done after considering a "number of potential tsunami scenarios. . ." Are they simulated? Are they collected?

Response (g):

We have simulated all the historical tsunami with different magnitudes, and the simulation results were collected into a database.

Comment (h):

Figure 3: This seems to be the location map of the study area. However the study area is not clear (scale is very small) and there is that green line and the epicenter. I would recommend to include a figure with much more detail of the study area and the other one (together with figure 4, for example) related to the travel time calculations.

Response (h):

We have modified this figure and added Fig.5 for the tsunami travel time.

Comment (i):

Lines 117-119: This is not a formal definition of evacuation time, but the interpretation of the authors. I miss either saying that the definition provided is how it is understood in this paper or the reference that has been taken.

Response (i):

This has been clarified in Lines 141-148 of the revised paper. Evacuation time is the available response time for evacuation (Post et al., 2009), less the tsunami travel time.

Comment (j):

Lines 117-130: The calculation performed to establish the tsunami travel times is not clearly described. I believe that the method should be replicable but any other person reading the document: for instance, How is the average water depth determined? Epicenter coordinates? Results (2 hours) may be good as a rough approximation but I do not believe they are comparable (at least in general terms) with numerical tsunami travel time models. The approximation will be acceptable depending also on the purpose followed. Maybe for evacuation purposes in a decision support system in a local area this is not the best. On the other hand, it could be debatable if average velocity is appropriate to estimate the time. . .

Response (j):

We have changed this method. We calculate the tsunami travel time using TTT in the revised paper.

Comment (k):

[Figure]

Lines 147-195: From this reviewer's point of view, vulnerability is not really being considered and is not properly defined. Maybe it could reshaped on some way explaining that some aspects that could influence in the vulnerability of the system are taken into account. . . There is confusion in the statement of lines 151-152. Paragraphs following the line 156 should follow the order detailed in the sentence, for an easier reading.

Response (k):

This has been clarified in lines 165-174. For vulnerability, we have analyzed population factors to specify the number of people at risk in the revised paper.

Comment (l):

Line 160 and Line 168: I really believe that elevation is not a preferred consideration for vulnerability analysis. It is a crucial parameter to establish the affected area but that is hazard and exposure.

Response (l):

The elevation is not a consideration for vulnerability analysis in the revised paper.

Comment (m):

Lines 186 to 189. Is the "8 km distance from shore" calculated based on the previous equation? This is not clearly stated.

Response (m):

The offshore distance is not a consideration for vulnerability analysis in the revised paper.

Comment (n):

Lines 192 to 195: It is true but has it been done in the frame of this work? If yes, please show results. If not, clarify it. Is really the vulnerability being calculated or estimated in some way? Is there any result achieve?

Response (n):

This has been deleted in the revised paper.

Comment (o):

Line 218: This is not very relevant when describing land use to estimate the impedance related to evacuation analysis

Response (o):

We have deleted this sentence.

Comment (p):

Line 224: There is no explanation of the figure.

Response (p):

This has been modified in lines 216-218 of revised paper.

Comment (q):

Lines 236-237: Congestion-prone roads are calculated based on population census and road classification data. Which is the approach followed? There is no explanation. It seems obvious than "worst roads" and most densely populated areas will have more congestion-prone roads, but it is no explained. Results are not explained either.

Response (q):

This has been modified in Lines 229-235. In our decision support system, congestion-prone roads were analyzed by overlying population census data and road classification data. Roads are easily congested in places where there is a larger populace.

Comment (r):

Lines 241-246: It is no clear if vertical shelters are added to the system based on calculations that are not the objective of this paper or if this have been addressed in this work. It is stated, as the only apparent reason, that evacuation shelters are usually se-
lected "following the principles of good accessibility and large capacity, among others".
This is a quite poor reasoning. As figure 10 has not been explained, the reader is not
able to understand how those green points are calculated. On the other hand, if there
has not been inundation modelling, which is the procedure followed to locate them?

Response (r):

This has been clarified in Lines 238-240. At the beginning of the evacuation analy-
sis, some shelters are provided by the local managers. New shelters can be added
according to the need in the system. We use the inundation model in the revised paper.

Please also note the supplement to this comment:
http://www.nat-hazards-earth-syst-sci-discuss.net/nhess-2016-319/nhess-2016-319-
AC1-supplement.pdf

―――――――――――――――

**Supplement:**

[revised manuscript text omitted]

---

## Author Comment (AC2) · 9 Feb 2017

Authors' response to Anonymous Referee #2

Firstly, we would like to thank this referee for accepting to review this paper and for the in-depth and positive evaluation and useful comments. We have carefully considered all the comments and made the suggested changes in our manuscript. Our responses to the comments are shown below. Our responses to the comments are shown below. The revised paper is in the supplement.

Referee #2 General comments:

Title of this manuscript sounds interesting but the detail seems to be lower than what have stated in the title. The manuscript needs major improvement in their method, study area, explanation and English writing so that it can be reached to international standard. Please find my comments as shown below. Major comments - Only one study area in China cannot represent the evacuation for whole SCS region. If the authors would like to do such quick and simple method, at least they should have one or two more target areas to show performance of the system in both near-field and far-field tsunami. Verification of the model is needed to convince readers that even such simple method is applicable. - How fast is the processing time required for this decision support system? I suggest to add information of time in Fig. 1. I believe that each country in SCS region has different timing of tsunami warning preparation. Please add information about this and tell readers how this system can help at each timing for SCS countries. - The use of English should be very much improved by a native speaker.

General Response:

We have changed the name of this paper to "Development of a decision support system for tsunami evacuation: application to the Jiyang District of Sanya City in China". We have modified the framework of this paper, adding the tsunami travel time. And this paper has been improved by a native English speaker. It may take 2 minutes to get tsunami hazard information using the decision support system. The vulnerability analysis requires 2 to 4 minutes. It takes 5 minutes for the system to provide tsunami static analysis results. Tsunami dynamic evacuation analysis may take several hours, but this analysis can be used to study local tsunami evacuation problems before a tsunami occurs.

Specific comments:

Comment (a):

Title: The title should be more specific by mentioning the study area. L36-37: This reference Benard (2005) is too old to support the statement "Currently, many governments."

Response (a):

The title of this paper has changed into "Development of a decision support system for tsunami evacuation: application to the Jiyang District of Sanya City in China". The reference Benard (2005) has been replaced by "Scheer S, Gardi A, Guillande R, et al. Handbook of Tsunami evacuation planning[J]. Retrieved March, 2011, 3: 2013."

Comment (b):

L53: Objective and purpose of this paper should be written at the end of section 1.

Response (b):

We have added the purpose of this paper in Lines 38-44 of the revised paper.

Comment (c):

L55-67: More literature review is needed. There are more types of evacuation model than what have mentioned in this part. I don't think that the agent-based model is only for evacuation drill. It can be for disaster planning as well. What do you mean by "the decision makes do not know well"?

Response (c):

More literature review has been added in Line 46 of the revised paper. Both the least-cost-distance model and the agent-based model are adopted in the revised paper.

Comment (d):

L70-78: Should other items in Fig. 1 be explained? How much detail of fault mechanism considered in the database? Do they also have events outside of subduction zone in the database? Explain briefly about evacuation cost here.

Response (d):

The framework has changed in Line 56. The detail of fault mechanism is in Lines 114-123.

Comment (e):

L117-129: Evacuation time can also be a time from a natural warning (i.e. ground shaking) in case of no warning system. The authors should scientifically state why equation (1) is proper to be applied to this region. They should also more clear by saying that the tsunami travel time is estimated by dividing the distance from epicenter to target area by tsunami celerity, C (not to be confused with flow velocity). What do you mean by "numerical tsunami travel time model"? Fig. 4 should be improved by adding the location names (i.e. China, the Philippines, Hinan, Luzon, Manila Trench) and the estimated tsunami celerity during the deeper (4,000 m) and shallower (2,000 m) section in the figure

Response (e):

We have modified the method of tsunami travel time calculation. The tsunami travel time is calculated by the numerical model TTT in the revised paper, not estimated by the water depth.

Comment (f):

L133-142: "Influence area", what is the meaning of "influence"? Is it from tsunami amplitude, arrival time or any other parameters? The simple radius based on earthquake magnitude shown in Fig. 5 is too simple. As the authors had mentioned in section 4.2 that the evacuation time is calculated by the sea depth, the sea depth, coastal topography of each communities for both sides of the Philippines are quite different and cannot be just simply represent by the radius.

Response (f):

This method has been modified in the revised paper. The tsunami hazard information is obtained from two numerical model COMCOT and TTT. The numerical calculation is conducted in advance, and the results are imported in to a database.

Comment (g):

L161: How do you relate "magnitude (do you mean earthquake magnitude?)" to "physical nature of tsunami"?

Response (g):

We have deleted this paragraph because the framework of the system was changed.

Comment (h):

L182-191: I am not sure if the 8.3 km was really happened. Where was it and what kind of topography? Please scientifically explain why equation (2) is suitable to apply to your study area or other areas in SCS region. Only tsunami height is a parameter for equation (2). Such large overestimation can be occurred in mountain areas where tsunami is limited by the topography. The authors should also explain why they used the distance of 8 km as shown in Fig. 8. What is their expected Y0 to get 8 km distance?

Response (h):

This paragraph has been deleted because the offshore distance is not used to analyze the vulnerability. We have modified the framework of the system.

Comment (i):

L221-237: There are many parameters related to the traffic during evacuation such as road width, traffic regulation during tsunami warning, distribution of evacuation shelters, ratio of evacuation using car, number of people in one car, day or nighttime that the authors did not mention. Explanations of the evacuation cost analysis is too simple and not enough to understand.

Response (i):

Some of these parameters are considered in the revised paper, such as road width, evacuation shelters, ratio of evacuation using car. The percentage of evacuation by car is from 0 to 100% in the dynamic evacuation analysis. The scenarios of day and nighttime are not considered in the system. Budiarjo (2006) estimate the population in the house by assuming 50% of the occupants are outside the house in day scenario. This needs more detail data about population and houses. Future research will focus on this. Explanations of the evacuation cost analysis is in lines 207-218.

Comment (j):

L247-262: The conclusion is rather simple. They should write, for example, major finding in their study or benefit (from their discussed results) to agent-based model or suggest how to develop your proposed system to other countries or regions.

Response (j):

The conclusion has been modified in Lines 285-311.

Please also note the supplement to this comment:
http://www.nat-hazards-earth-syst-sci-discuss.net/nhess-2016-319/nhess-2016-319-AC2-supplement.pdf

**Supplement:**

[revised manuscript text omitted]